# A Cross-Sectional Survey of Musculoskeletal Injuries in South African Shotokan Karate

**DOI:** 10.3390/jfmk10040463

**Published:** 2025-11-27

**Authors:** Mikala de Wet, Christopher Yelverton

**Affiliations:** Department of Chiropractic, Faculty of Health Sciences, University of Johannesburg, Johannesburg 2092, South Africa; mikaladewet4@gmail.com

**Keywords:** combat sport, injury epidemiology, martial arts, Shotokan karate, athletic injuries

## Abstract

**Objectives:** This study investigated the prevalence and severity of musculoskeletal injuries within South Africa’s most popular karate style, Shotokan, a previously unexamined area. As an exploratory study, it aimed to generate hypotheses by determining the prevalence, severity, and nature of these injuries to address this significant gap in the national combat sports literature. **Methods:** A descriptive, cross-sectional design was employed, utilizing a confidential online questionnaire distributed through various Shotokan organizations. The study gathered 155 responses (26.85% response rate). **Results:** The findings revealed a high injury prevalence, with 47.3% of participants reporting at least four injuries. These injuries occurred equally in training and competition (56.5%) and developed both acutely and over time (53.4%). Experienced practitioners at the Shodan level were particularly affected. The knee was the most frequently injured body part (11.6%), and muscle strains were the most common injury type (19.3%). Notably, 26.2% of karatekas continued training despite being injured. A significant weak positive correlation was found between years of training experience and injury levels (rs = 0.275, *p* = 0.007). However, no significant associations were found between injury prevalence and age, BMI, or training frequency. General practitioners were the most consulted healthcare professionals (22.0%). **Conclusions:** This study establishes a high prevalence of musculoskeletal injuries among South African Shotokan karatekas, particularly associated with experienced practitioners. These findings are hypothesis-generating, and the cross-sectional design precludes causal inferences. The data provides a crucial foundation for future longitudinal research to investigate causality and for developing evidence-based injury prevention protocols, particularly for the knee.

## 1. Introduction

Martial arts can be defined as any art of combat or method of self-defense and describes many forms of combat sports [1]. Most forms of martial arts stem from self-defense methods that have developed into sports. Some popular forms of martial arts from Japan include sumo, judo and karate [2]. Other popular martial arts include Muay Thai, Taekwondo, Tai Chi, Kickboxing, Kung Fu, Jiu-Jitsu and Krav Maga [3], each form having its own history and origin, focusing on different aspects such as grappling modalities involving throws, immobilizations and joint locks, and impact modalities involving punches, kicks, and elbow and knee blows [4].

Previous research on musculoskeletal injuries in martial arts has demonstrated that the injury rate is dependent on the rules allowed within the particular form, such as if full contact is accepted or if it is a semi-contact sport, with full contact resulting in the injury rate being much higher compared to semi-contact [2].

Grappling forms of martial arts, like Judo, have more joint injuries compared to striking forms, like karate, which result in more muscle injuries. A difference in injury patterns was also noted depending on the individual’s level of experience in the sport. Injury levels were dependent on factors such as grappling or impact modalities, experience level, fitness and conditioning level, age, and use of protective equipment, as well as whether it was a training or competitive environment [4].

The majority of studies investigating musculoskeletal injury prevalence and severity amongst karate practitioners are based on injuries sustained during a particular event and do not consider injuries sustained over time during training [5]. These studies are not based on a South African context and do not look at a particular style of karate but rather a combination of styles. Shotokan karate is the most widely practiced style of karate in South Africa, with more than fifteen large and established associations and thousands of active participants. Its distinct techniques and movement patterns present unique injury risks, making research into this style important for effective injury surveillance and the development of prevention strategies [6]. Despite its prevalence, there is limited published research on Shotokan karate in the South African context. This lack of data may be attributed to the longer-established karate research communities, greater sports research funding, and larger karate associations present in countries abroad [7]. Conducting research locally is essential for identifying context-specific injury patterns and risk factors, while also considering cultural and demographic differences that may influence participation and injury rates. Previous systematic reviews have indicated that more studies are required to determine the effect of BMI, experience levels and age on musculoskeletal injuries sustained in karate [8].

The primary aim of the exploratory study was to determine the prevalence and severity of musculoskeletal injuries in Shotokan karate in South Africa, and secondarily to determine the association between the age group, BMI group, belt ranking, and years of experience with musculoskeletal injuries. Understanding these patterns is crucial for developing evidence-based injury prevention strategies. For instance, identifying associations with factors like BMI, years of experience, or specific training volumes can inform targeted interventions, such as tailored conditioning programs for at-risk athletes or guideline development for coaches to enhance safety during training and competition.

## 2. Methodology

This study utilized a descriptive, cross-sectional design with a non-probability, voluntary response sampling method.

We employed QuestionPro survey software (version 5.7.4) with initial questions used to obtain consent from the participants. Accompanying the survey was an online information letter, which indicated the study’s purpose, ethical considerations and the primary researchers’ contact details. The study was confidential, voluntary and anonymous; participants could withdraw their responses up to the point of submission of the survey. Inclusion criteria were individuals over 18 years old who practice Shotokan karate for a minimum of 6 months, of any belt ranking. The survey was opened on 22 April 2025 and closed on 5 June 2025.

The survey (see Appendix A) was distributed with the assistance of eight major Shotokan karate organizations in South Africa. A total of 36 individual dojos (clubs) affiliated with these organizations agreed to participate. Based on communication with dojo instructors, the average club size was estimated to be 30 members. However, as this was an aggregate estimate provided by organization heads and not based on a full census of individual dojo membership rolls, a standard deviation or precise range cannot be provided. This estimate should be interpreted with this limitation in mind. Given that the inclusion criteria required participants to be aged 18 years or older, it was estimated that approximately 40% of any given club’s membership (approximately 12 members per club) would be eligible for the study.

Therefore, the total estimated accessible population (*N*) was calculated as follows: 36 clubs × 12 eligible members per club = 432 individuals.

Using a sample size calculator with a confidence level of 95% and a margin of error of 5%, the recommended minimum sample size was 205 participants. With an expected response rate of 50% for online surveys distributed through community channels, a target of 110 responses was deemed acceptable to exceed the minimum threshold.

The survey link was distributed by the head instructor of each participating dojo to their eligible adult members.

The survey’s questions were adapted from studies [8,9,10,11,12,13], which were also related to musculoskeletal injuries (for the purposes of this study being defined as any physical complaint sustained during karate training or competition that resulted in at least one complete day of missed training or competition) in martial arts and included questions about the participants’ demographics, karate information and karate injury information. It also included pre-set questions from the International Physical Activity Questionnaire (IPAQ). A pilot study was conducted prior to the release of the questionnaire to the participants, whereby the survey was sent to five Shotokan karate students, who were excluded from the final study. Informal feedback was provided, highlighting minor phrasing issues in two questions and the addition of “ribs” under the anatomical areas section of a particular question. Adaptations to the survey were made accordingly, and the feedback improved content validity.

The questionnaire consisted of 31 questions and four subsections. Section A included patient demographics and questions regarding the patient’s age, height, weight and gender; the patient’s BMI was calculated using the height and weight measurements [12]. Section B included pre-set questions from the short form of the International Physical Activity Questionnaire (IPAQ) [14]. Section C included questions for the participants’ karate background information, which was related to the frequency and amount of training, belt ranking, and years of experience in Shotokan karate [8,9]. Section D was questions related to karate injuries [8,10,11], and these questions were specific to location, type and extent of injury, based on number of days that have elapsed from the date of injury to the date of the player’s return and categorized as a minimal injury defined as >24 h—3 days of activity missed, mild (4–7 days), moderate (1–4 weeks) or severe (4+ weeks) [15], as well as healthcare professionals’ preferred treatment for the injuries.

With assistance from Statistical Consultation Service (Statkon), a descriptive statistical analysis was performed, using SPSS (version 15). The counts, percentages, means, medians and standard deviations (SD) were calculated, and crosstabulations were used to compare the variables of participants’ BMI, injury occurrence, amount of training, body region and type of injury. For 2 × 2 tables, Fisher’s Exact test was performed; for tables larger than 2 × 2, the Pearson Chi-Square Test was used (a significant Chi-square result (*p* < 0.05) indicates that the two variables are not independent and that a statistically significant association exists between them). The *p*-values were calculated to determine the statistical significance (with significance set at 0.05 and a 95% confidence interval) between different variables in the cross-tabulations, and the Cramer’s V values were calculated to measure the strength between two categorical variables. A post hoc sensitivity analysis was conducted using GPower (version 3.1.9.7) to assess the statistical power of this study. Given the sample size (*N* = 155), an alpha level of 0.05, and the observed effect sizes, the analysis indicated that the study achieved a power of 0.72 to detect the primary association (between years of experience and injury levels). While this was sufficient to identify the significant association reported, it is below the conventional threshold of 0.80. This lower power means the study had a reduced sensitivity to detect smaller, yet potentially meaningful, effects.

Anonymity was maintained throughout the study, as no identifying information was collected from participants. There were no anticipated risks or direct benefits to participants, and no conflicts of interest were reported. Ethical approval for the study was granted by the Faculty of Health Sciences Research Ethics Committee (REC) at the University of Johannesburg (REC 241112-035). All analyzed data will be securely stored on the University of Johannesburg’s password-protected server.

## 3. Results

### 3.1. Demographic Data

Eight organizations were approached, representing an estimated 87 dojos nationwide. From these, 36 dojos formally agreed to participate. Instructors from these dojos distributed the survey to their adult members. The survey was open for 12 weeks. A total of 161 responses were obtained, and the response rate was thus 26.85%. As indicated in Table 1, the sex distribution of the survey indicated 63.23% (*n* = 98) were male participants, and 40.00% (*n* = 62) of participants were aged between 26 and 45 years old. The majority of BMIs (height and age calculation) indicated a median BMI of 26.77, which classifies as “overweight” those with a BMI between 25 and 29 kg/m^2^ (38.3%, *n* = 59).

### 3.2. Training Information

Years of training ranged from less than 5 years (18.06%, *n* = 28) to 16 years or longer (46.45%, *n* = 72). The most common frequency of training was twice a week (38.71%, *n* = 60), and the least common frequency was once per week (1.29%, *n* = 2). The length per training session was most commonly between 1 and 1.5 h (54.19%, *n* = 84), with the least amount of time being less than 30 min (0.65%, *n* = 1) and the most time spent more than 2 h in training (1.94%, *n* = 3) (Table 2).

### 3.3. Injury Information

Of the participants who reported injuries, 47.3% (*n* = 62/131) indicated that they sustained four or more injuries from karate. It is important to note that these figures represent the participants’ cumulative count of injuries over their karate practice, not distinct injury events from a specific period. The majority of these injuries were reported at the 1st dan black belt (Shodan) level (21.53% of all 288 reported injury instances, *n* = 62). The injuries were sustained both during training and competition (56.5%, *n* = 74), and 36.6% of participants sustained acute injuries (*n* = 48), whilst a lower 9.9% (*n* = 13) sustained the injuries over time. The anatomical area of most sustained injuries was the knee region, with 43.5% (*n* = 57) of participants, and buttock injuries were the least likely to occur (0.8%, *n* = 1) (Figure 1). Muscle injuries (strains) were the most common type of injury to occur (57.3%, *n* = 75), whilst loss of consciousness had the least occurrence (9.9%, *n* = 13). Over a quarter of participants (26.2%, *n* = 34) reported training despite their injuries, while a small minority (2.3%, *n* = 3) were unable to train for six months to a year or longer (Figure 2). When graded by time loss, the majority of injuries that required time off were classified as Severe (30.77%, *n* = 40), followed by Moderate (22.31%, *n* = 29) and Mild (12.31%, *n* = 16). The majority of participants indicated that they were treated by a general practitioner (GP) for their injuries (22%, *n* = 68), whilst 2.6% (*n* = 8) of participants were treated by other medical professionals, such as dentists, maxillofacial surgeons and orthopedic surgeons (Figure 2). The extent of injury for the participant’s most severe injury was that they required surgery (30.1%, *n* = 40), whilst 0.8% (*n* = 1) of participants indicated that they required a cortisone injection for their most severe injury (Table 3).

### 3.4. Associations Between Participant Characteristics and Injury Level

There was no significant association between age and number of injuries (*p* = 0.221, X^2^ = 3.02, df = 2). The association between activity level and number of injuries were statistically insignificant as well (*p* = 0.069, X^2^ = 5.35, df = 2). The association between BMI and the number of injuries had a statistically insignificant result (*p* = 0.327, X^2^ = 2.23, df = 2). A significant association was found between years of experience and number of injuries (*p* = 0.007, X^2^ = 9.94 and df= 2), with a weak effect size (Cramér’s V = 0.275). The association between frequency of training (days per week) and number of injuries had a statistically insignificant result (*p* = 0.314, X^2^ = 2.32 and df = 2).

## 4. Discussion

This study examined the prevalence, amount, severity, type and body region of musculoskeletal injuries in Shotokan karate in South Africa. This study found no significant association between BMI and injury prevalence, but a significant association was found between increased years of experience in training Shotokan karate and reported numbers of injuries. There was no link between frequency and duration of training and the rate of injuries found, and most injuries were sustained both during competition and training, rather than at one specifically. The knee was the most frequently injured anatomical body part, and muscle strains were the most common type of injury. Most participants continued training despite their injuries, and general practitioners were the most frequently consulted healthcare professionals.

### 4.1. BMI and Association with Injuries

While the association between BMI and injuries is complex, previous studies suggested that an increased body mass index may increase the association with musculoskeletal injuries due to the increased joint loads and greater force in kicks or strikes [16]. However, in this study, the relationship between BMI and prevalence of injuries was not significant. This could be due to the BMI measurement not differentiating between muscle mass and fat mass; BMI does not accurately report the functional stability and strength of muscles compared to the distribution of fat [17]. The increased amount of muscle mass could contribute to decreasing the prevalence of injuries since joints are better supported during movements, whereas the increased fat distribution will do the opposite; the joints will be less supported and thus more prone to injuries [17]. These findings suggest that BMI is not an adequate screening tool for association to injury of Shotokan karatekas in South Africa. BMI in athletic populations may be considered a more general anthropometric descriptor and may not be a valid indicator of body composition in muscular athletes. Future studies should use more precise measures like body fat percentage.

### 4.2. Years of Experience and Association to Injuries

In alignment with previous studies [10,18], an association was found when exploring the relationship between years of experience in Shotokan karate training and the number of musculoskeletal injuries. In this study, participants with 16 years or more of experience in Shotokan karate training reported a higher number of injuries compared to participants with less than 5 years of Shotokan karate training. This association could be explained by the cumulative exposure to repetitive movement over time, which may lead to overuse injuries [19]; however, this cross-sectional study cannot confirm a causal mechanism. This suggests a trend that increased years of experience in Shotokan karate in South Africa were associated with increased injury levels and may warrant a future prospective cohort study.

### 4.3. Duration and Frequency of Training

The majority of Shotokan karatekas in South Africa reported training in karate twice a week for between 1 and 1.5 h per session. There was no significant association between the number of injuries and the duration and frequency of training. An increased duration and frequency of training have been proposed to increase the association of musculoskeletal injury [10]; however, this will be affected by the type and intensity of training performed at the specific club. More research is needed to look at injuries during training, as the majority of studies explore injuries at competitions [5]. For South Africa’s Shotokan karate athletes, the findings indicate that the variation in training intensity and frequency may differ across clubs, and they may not all have the same trend in injuries.

### 4.4. Injuries During Training or Competitions

Contrary to previous studies [10,20], while many participants indicated that their injuries were sustained during both training and competitions, more participants indicated that their injuries were sustained during training sessions. This could be attributed to the repetitive movements performed during training, as well as the increased time and frequency of training compared to the short bouts and decreased frequency in competitions [21]. Many previous studies only examined the injuries reported at competitions and did not include injuries occurring during regular training sessions [5]. For Shotokan karatekas in South Africa, these findings could indicate that a higher percentage of injuries occur during training.

### 4.5. Acute or Chronic Nature of Injuries

Injuries appear to develop both acutely and over time; a greater proportion of the participants indicated that their injuries were sustained acutely rather than chronically. Acute injuries in karate are predominantly associated with the high speed and competitive nature of the sport [22]; however, only a small number experienced more chronic injuries, which may reflect the repetitive nature of the movements characteristic of Shotokan karate [5]. These findings could suggest that South African Shotokan karatekas should maintain the practice of proper techniques and appropriate warm-up and cool-down routines to minimize the potential risk of acute injuries [23].

### 4.6. Anatomical Regions of Injuries

The knee was the most frequently reported anatomical site of injury in this study. A high prevalence of knee injuries may be attributed to biomechanical stresses, rotational forces, and the distinctive deep stances and explosive techniques characteristic of Shotokan karate [23]. The knee was the most frequently injured body part (11.6%), a finding consistent with studies in Taekwondo (15.2%) and Brazilian Jiu-Jitsu (10.5%) [3]. This commonality across striking and grappling arts suggests that the dynamic, pivoting movements fundamental to martial arts place universal stress on the knee’s stabilizing structures. The second most commonly injured region was the head/face, consistent with previous findings [16,23,24]. The increased prevalence of head and facial injuries may be explained by direct contact during kumite (sparring) or the execution of traditional techniques during partner training. Given that the majority of participants were aged between 25 and 34 years, it is likely that many competed in senior divisions, where semi-contact to the face is permitted; thus, the risk of head and facial injuries is elevated [16]. These findings suggest that Shotokan karatekas in SA should focus on applying the appropriate techniques when performing certain movements, which may put the knees at risk of injuries, such as practicing the correct methods of kicking and stances [16].

### 4.7. Types of Injuries

Consistent with previous research [25], this study identified muscle injuries, specifically sprains, as the most commonly reported injury type among participants. This may be attributed to the biomechanical demands of Shotokan karate, particularly the use of deep stances and explosive techniques, which place considerable stress on muscles through rapid contraction and relaxation cycles [26]. In contrast, contusions were the third most frequently reported injury type in this study. This differs from previous studies [10,16,20], which identified contusions as the most prevalent injury type. A possible explanation for this discrepancy is that participants in the present study were asked to report their most severe injury, which may have resulted in less severe injuries—such as minor contusions—being underreported. With muscle strains being the most commonly reported injury, South African Shotokan karatekas should practice appropriate warm-up routines to minimize the potential of muscle strains when practicing karate [23].

### 4.8. Protective Equipment and Injuries

The use of personal protective equipment—such as gloves (mitts), shin and foot protectors, mouth guards, groin guards, and breast/chest protectors—has been suggested to reduce the impact of contact and potentially lower the prevalence of injuries in Shotokan karate [16]. However, the present study found no difference in injury occurrence between participants who wore protective equipment and those who did not. This finding may be attributed to variations in the type, quality, and consistency of protective equipment usage across Shotokan dojos in South Africa. The inconsistent usage of protective equipment during karate training could be attributed to the prevalence of injuries reported, as the equipment is not worn often enough to make an association with the reported injuries [5]. Additionally, it is possible that practitioners may consciously reduce the level of contact when not wearing protective equipment, thereby mitigating potential injury regardless of gear use [5,16]. Although the findings of this study were not consistent with other studies, there are several factors that could have influenced the findings, as mentioned. South African Shotokan karatekas are still encouraged to practice the appropriate usage of personal protective equipment to minimize the potential for sustaining musculoskeletal injuries whilst practicing this semi-contact sport [23].

### 4.9. Recovery Time Period Post-Injury

Only about a quarter of participants continued training despite their injuries and required no recovery period or took two to three months off for recovery. These results align with previous studies [10,20,23,27], which found that most Shotokan karate injuries are minor, require short recovery times, and are often managed without formal medical care. Thus, South African Shotokan karatekas are encouraged to take note of their injuries and seek professional help if appropriate.

### 4.10. Extent of Injury Treatment

Around a third of participants reported requiring surgery for their most severe injury, with a quarter requiring treatment and rehabilitation by a healthcare professional. These findings support the conclusions that moderate to severe injuries often necessitate medical intervention and can have a significant impact on an individual’s functioning and quality of life. While the majority of reported injuries were minor, the finding that 30% of participants required surgical intervention for their most severe injury presents a complex picture. This suggests that while the frequency of minor injuries aligns with a semi-contact sport, the potential for serious injury is non-trivial. Therefore, the risk profile of Shotokan karate may be best characterized not as universally ‘safe,’ but as one with a high volume of low-severity injuries and a significant minority of high-severity events requiring advanced medical care. The majority of injuries reported in this study were minor; the extent of treatment amongst South African Shotokan karatekas varied considerably based on injury severity.

### 4.11. Healthcare Professionals’ Treatment

Physiotherapists and biokineticists have been indicated as the most consulted healthcare professionals for sports injuries [6]; however, the most frequent care in our study was from general practitioners, potentially a reflection of their accessibility in South Africa [28]. Self-treatment—including rest, ice, and over-the-counter medication—was the most frequent, supporting earlier findings that most injuries were minor and required short recovery [23]. South African Shotokan karate instructors are encouraged to educate their students on the appropriate healthcare professionals available for consultation in the event of an injury [23].

### 4.12. Implications for Injury Prevention and Future Research

A potential consideration for the high prevalence of knee injuries and muscle strains identified in this cohort may be proposed as a potential deficit in dynamic joint stability. The critical role of postural control and neuromuscular stability in preventing musculoskeletal injuries has been previously presented [29]. Our findings, combined with this evidence, suggest that integrating balance and proprioceptive training, such as the single-leg stance protocols, may be beneficial in karate conditioning to mitigate common injuries. Future prospective studies should consider investigating the efficacy of such neuromuscular training programs in reducing injury incidence among karatekas.

### 4.13. Limitations

This study has several limitations that should be considered when interpreting the results. First, its cross-sectional design precludes any causal inferences and lacks temporal control; the association between greater experience and higher injury reporting, for instance, is likely a function of accumulated exposure over time rather than a direct causal relationship.

Methodologically, the reliance on self-reported data from an online survey introduces potential biases. Recall bias may have affected the accuracy of injury details, particularly for “most severe injury” reporting. Furthermore, selection bias is probable, as the survey was distributed to active dojo members, excluding former practitioners whose careers may have ended by injury, and potentially leading to an underestimation of severe outcomes. The non-probability sampling and a final response rate of 26.85% further limit the generalizability of the findings.

Key methodological shortcomings also affect data interpretation. The absence of a defined recall period for injuries makes it difficult to distinguish between point and period prevalence and hinders direct comparison with other epidemiological studies. The lack of objective medical verification and a standardized injury severity scale (e.g., based on time loss) means that injury types and their clinical impact may be misclassified, limiting comparability. Additionally, the study did not account for participation in other sports, meaning the reported injury prevalence may reflect a cumulative trauma burden not exclusively attributable to karate.

Finally, analytical constraints must be acknowledged. A post hoc sensitivity analysis revealed the study was underpowered (power = 0.72) to detect weak effect sizes. Therefore, the non-significant findings for factors such as age, BMI, and training frequency should be interpreted with caution, as the study may have lacked the sensitivity to identify these more subtle relationships. The scope was also limited to physical and demographic variables, omitting psychological and neuromuscular factors, which are critical for a comprehensive understanding of rehabilitation and recurrence, as highlighted in contemporary sports medicine frameworks.

## 5. Conclusions

This study identified a high prevalence of acute knee and head/face injuries among South African Shotokan karate practitioners, with reporting particularly associated with experienced and Shodan-level athletes. Most injuries occurred during training, were minor, and required little or no recovery time, though severe cases—primarily muscle sprains—were usually treated by general practitioners. No significant association was found between BMI, training frequency or duration, and protective equipment use did not reduce injury occurrence.

These findings have revealed the prevalence and severity of musculoskeletal injuries experienced in South Africa amongst Shotokan karate participants. The findings suggest that injury association in this semi-contact sport is more strongly influenced by variations in training quality and methods than by athlete characteristics or equipment use. The injury profile reveals a high volume of minor injuries, consistent with a semi-contact sport, alongside a significant minority of severe injuries. This underscores the need for continued safety initiatives despite the generally low severity of most reported incidents. Future research should employ larger, longitudinal studies to track injury trends and severity over time.

## Figures and Tables

**Figure 1 jfmk-10-00463-f001:**
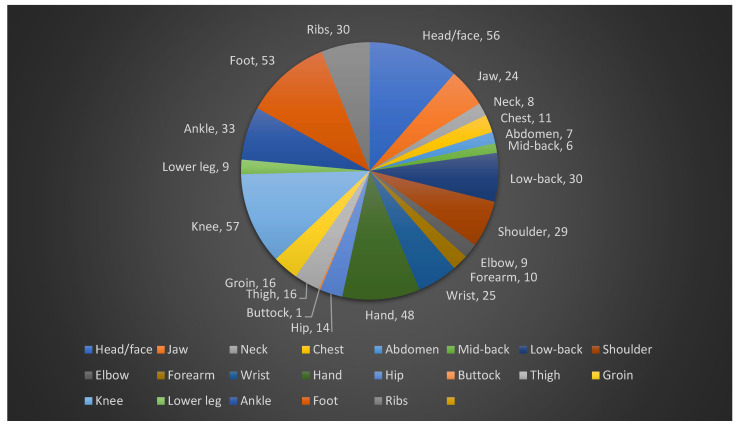
Pie chart illustrating the anatomical region of injury.

**Figure 2 jfmk-10-00463-f002:**
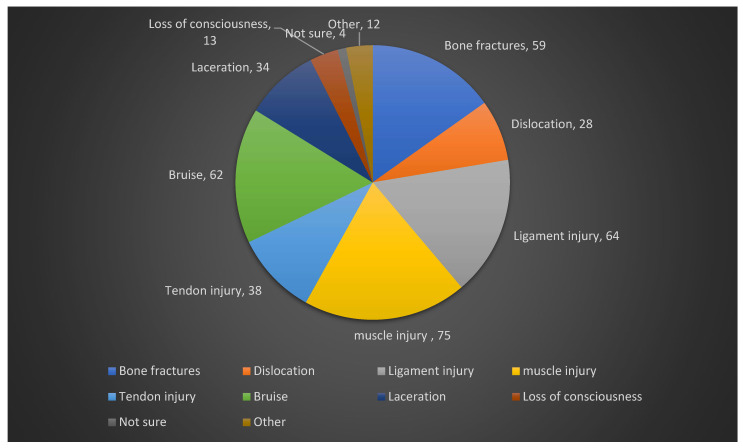
Pie chart illustrating types of injury.

**Table 1 jfmk-10-00463-t001:** Demographic data, indicating age, height, weight and calculated BMI.

Characteristic	% (*n*)	Mean (SD)
**Age (years)**		18.53 (4.85)
18–24	21.94 (34)	
25–34	22.58 (35)	
35–44	18.06 (28)	
45–54	19.35 (30)	
55–64	10.32 (16)	
≥65	7.10 (12)	
**Sex**		
Male	63.23 (98)	
Female	36.13 (56)	
Prefer not to say	0.65 (1)	
**Height (cm)**		
130–139	0.65 (1)	
140–149	0.65 (1)	
150–159	9.09 (14)	
160–169	22.73 (35)	
170–179	40.91 (63)	
180–189	24.03 (33)	
>190	1.95 (3)	
**Weight (kg)**		
42–54	3.23 (5)	
55–64	16.13 (25)	
65–74	18.06 (28)	
75–84	22.58 (35)	
85–94	20.00 (31)	
95–115	14.19 (22)	
116–125	3.23 (5)	
>126	2.58 (4)	
**BMI (kg/m^2^)**		27.08 (5.26)
<18.5	1.30 (2)	
18.5–24.9	35.06 (54)	
25–29.9	38.31 (59)	
30.00–34.99	16.23 (25)	
35.00–39.00	7.14 (11)	
≥40.00	1.95 (3)	

**Table 2 jfmk-10-00463-t002:** Karate training information, including years of training, frequency and duration of training sessions.

Characteristic	% (*n*)
**Years of training**	
<1	3.23 (5)
1–2	3.23 (5)
3–5	11.61 (18)
6–10	16.77 (26)
11–15	18.71 (29)
16–20	12.90 (20)
>20	33.55 (52)
**Training session duration (minutes)**	
<30 min	0.65 (1)
30–60 min	32.26 (50)
1–1.5 h	54.19 (84)
1.5–2 h	10.97 (17)
>2 h	1.94 (3)
**Days trained per week**	
1	1.29 (2)
2	38.71 (60)
3	28.39 (44)
4	16.13 (25)
5	11.61 (18)
6	3.87 (6)
7	0 (0)

**Table 3 jfmk-10-00463-t003:** Participants’ injury information related to the number of injuries, belt ranking when injury was sustained, period of injury, acute/chronic injury, protective equipment worn, duration unable to train, healthcare professionals attended, extent of treatment.

Characteristic	%(*n*)	Median (IQR)
**Number of injuries**		>4 injuries (2 to >4)
1	12.98 (17)	
2	20.61 (27)	
3	19.08 (25)	
>4	47.33 (62)	
**Belt ranking when the injury was sustained**		1st dan black belt (Brown 1 to 2nd dan)
White belt	1.39% (4)	
Yellow belt	2.44 (7)	
Orange belt	1.74 (5)	
Green belt	1.74 (5)	
Blue belt	3.47 (10)	
Purple belt	2.78 (8)	
Red belt	3.47 (10)	
Brown belt 1	6.60 (19)	
Brown belt 2	4.86 (14)	
Brown belt 3	10.07 (21)	
1st dan black belt	21.53 (62)	
2nd dan black belt	15.63 (45)	
3rd dan black belt	10.76 (31)	
4th dan black belt	6.25 (18)	
5th dan black belt	4.17 (12)	
6th dan black belt	2.43 (7)	
7th dan black belt	0.69 (2)	
**Period of injury**		
Tournament/competition	8.40 (11)	
Training	35.11 (46)	
Both	56.49 (74)	
**Acute or chronic injury**		
Immediately	36.64 (48)	
Over-time	9.92 (13)	
Both	53.44 (70)	
**Protective equipment worn**		
Gloves	22.70 (69)	
GI	28.29 (86)	
Shin protectors	4.93 (15)	
Foot protectors	4.93 (15)	
Chest protectors	4.28 (13)	
Breast protectors	2.63 (8)	
Gum guard	20.72 (63)	
Face mask	0.33 (1)	
Groin guard	0.99 (3)	
Did not wear any equipment	10.20 (31)	
**Duration unable to train (injury severity grade)**		3 weeks (1–2 weeks to 2–3 months)
I trained with all my injuries (not graded)	26.15 (34)	
A few days (minimal)	8.49 (11)	
1–2 weeks (mild)	12.31 (16)	
3 weeks (moderate)	12.31 (16)	
1 month (moderate)	10.00 (13)	
2–3 months (severe)	19.23 (25)	
4–6 months (severe)	6.92 (9)	
6 months–1 year (severe)	2.31 (3)	
>1 year (severe)	2.31 (3)	
**Healthcare professionals attended**		
General practitioner	22.01 (68)	
Chiropractor	8.74 (27)	
Physiotherapist	18.77 (58)	
Biokinetics	9.06 (28)	
Orthopedic surgeon	14.24 (44)	
I treated myself (rest, ice, medication, etc.)	20.71 (64)	
I did not receive any treatment	3.88 (12)	
Other	2.59 (8)	
**Extent of treatment**		
Required no treatment	8.40 (11)	
I treated myself	11.45 (15)	
Diagnosed and sent home to recover	18.32 (24)	
Required treatment and rehabilitation by a healthcare professional	26.72 (35)	
Hospitalized for injury	3.82 (5)	
Required surgery for injury	30.53 (40)	
Other	0.76 (1)	

## Data Availability

The original contributions presented in this study are included in the article. Further inquiries can be directed to the corresponding author.

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
