# Peer review of "J. Funct. Morphol. Kinesiol.2025, 10(4), 463;https://doi.org/10.3390/jfmk10040463"

_jfmk, 2025, doi:10.3390/jfmk10040463_

Round 1

Reviewer 1 Report

Comments and Suggestions for Authors

General:

Thanks for the opportunity to review this manuscript. I generally don’t rank survey-based studies particularly high. However, this one is well done, and is appropriate. I will leave it up to the editor to choose if a survey-based study ranks high enough for inclusion in the journal.

Title:

The title reads well, but I think it should be clear that the study is based on survey results. Please alter the title accordingly.

Abstract:

The p-value should be accompanied by a correlation statistic (possibly Spearman’s rho?) when reporting the association between training experience and injury occurrences.

Otherwise, the abstract is clear and nicely written, while highlighting the key methods, results, and takeaways.

Introduction:

While the introduction is strong and well-written, I believe it can be improved by explaining slightly why the gathered information is important/how it can be used.

For example, perhaps knowing if BMI is associated with injuries might encourage coaches and athletes to keep BMI within a specific range to potentially avoid injuries. I’m sure there are several other potential reasons for knowing the information surveyed.

Typically, a hypothesis will follow the aim statement, although I can understand if the authors choose not to include one based on the exploratory nature of this study.

Methods:

The sentence “The survey used QuestionPro to collect responses and had consent questions at the start of the survey to indicate the participants' consent.” Should be re-worded to read something like ‘We employed QuestionPro survey software with initial questions used to obtain consent from the participants.’

It looks like there are two spaces between the sentences on line 80. Please check.

Might the authors be able to provide a standard deviation, or range, for the estimated average club side of 30?

I believe that the exact survey question should be provided via either a link to an electronic server, or as a supplementary table.

Results:

The tables do a nice job of summarizing the results, and the writing is once again solid.

While the angled view of the pie-charts looks nice, they do make the ‘front’ sections look relatively larger than the ‘back’ sections. Thus, I recommend a more straight-on view.

It is not clear how the X2 statistics should be interpreted. The authors should add interpretation information in the methods (or results if they believe that to be more appropriate).

Discussion:

The discussion is nicely done, and includes some explanation of how the results can be interpreted (what change in AA might be associated with what change in HRV etc.). This is a great thing, but might be better in the results section (even just an example or two)?

Great job with the discussion. My only suggestion is to include statistics from the reference studies rather than simply stating the main findings. This might help to build additional context for the reader.

Reviewer 2 Report

Comments and Suggestions for Authors

Major Concerns

Study Design and Methodology

  • The cross-sectional, descriptive design based solely on self-reported online data does not allow for causal inferences between exposure variables (e.g., experience level) and injury risk.

  • The non-probabilistic voluntary response sampling introduces substantial selection bias, as individuals more affected by injuries may have been more motivated to participate.

  • The low response rate (26.85%) further undermines external validity, especially given that the estimated population size was derived from unverified dojo membership figures.

  • The survey instrument was adapted from previous studies but not fully validated for the local Shotokan population, which raises concerns about content and construct validity.

  • Data were collected over an unspecified recall period, which affects the interpretation of “prevalence” versus “incidence.”

Measurement of Variables

  • The definition of “injury” is not clearly operationalised (e.g., medical confirmation, time-loss criteria, or severity grading).

  • The absence of a standardised injury severity scale limits comparability with other epidemiological studies in martial arts.

  • The use of BMI as a risk factor is questionable, as BMI does not distinguish between fat and muscle mass — an important limitation in athletic populations.

  • The study does not account for participation in other sports, which could confound injury data.

Statistical Analysis

  • The analysis relies solely on bivariate comparisons (Chi-square and Fisher’s exact tests), without any multivariable modelling to adjust for potential confounders.

  • The Cramer’s V value of 0.275, described as a “weak-to-moderate effect,” should correctly be classified as weak, and the clinical significance of the p-value (p=0.007) is limited.

  • No power analysis or post-hoc sensitivity test was performed to assess whether the sample size was adequate to detect meaningful effects.

Presentation of Results

  • Tables and figures are overly dense with descriptive percentages, yet interpretation and synthesis are limited.

  • The pie charts are visually cluttered, reducing clarity and interpretability.

  • Certain numeric inconsistencies appear, e.g., the statement that “47.3% of participants reported at least four injuries” lacks clarification on whether these represent distinct or cumulative events.

Discussion

  • The Discussion is largely descriptive, reiterating results rather than critically interpreting them in light of the literature.

  • There is limited quantitative comparison with prior studies on karate or other martial arts styles.

  • The conclusion that Shotokan karate is a “safe sport” is inconsistent with the reported data — particularly the 30% of participants who required surgery for their most severe injury.

  • The authors at times imply causality (e.g., “years of experience increased injury risk”), which is not supported by the study design.

  • The Discussion could be strengthened by integrating findings from recent research on postural control and rehabilitation, such as “Assessment of Balance During a Single-Limb Stance Task in Healthy Adults: A Cross-Sectional Study” (DOI: 10.1177/00315125241277250), which highlights the role of neuromuscular stability in injury prevention.

Limitations

  • Recall and selection bias are appropriately mentioned, but other limitations are insufficiently explored, such as the lack of temporal control, absence of objective injury verification, and restriction to active dojo members.

  • The authors acknowledge that they measured prevalence rather than incidence, yet the recall window remains undefined, making comparisons across studies problematic.

  • The omission of psychological or neuromuscular recovery variables prevents a comprehensive interpretation of injury recurrence and rehabilitation. In this regard, integrating the framework proposed in “Unlocking the Power of Muscle Memory: Advanced Techniques for Post-Traumatic Rehabilitation and Return to Competitive Sports” (DOI: 10.25259/JMSR_261_2024) could provide valuable insight into long-term functional outcomes.

Round 2

Reviewer 2 Report

Comments and Suggestions for Authors

Thank you